# SEQUENTIAL ENUMERATION IN LARGE LANGUAGE MODELS

## ABSTRACT

Reliably counting and generating sequences of items remain a significant challenge for neural networks, including Large Language Models (LLMs). Indeed, although this capability is readily handled by rule-based symbolic systems based on serial computation, learning to systematically deploy counting procedures is difficult for neural models, which should acquire these skills through learning. Previous research has demonstrated that recurrent architectures can only approximately track and enumerate sequences of events, and it remains unclear whether modern deep learning systems, including LLMs, can deploy systematic counting procedures over sequences of discrete symbols. This paper aims to fill this gap by investigating the sequential enumeration abilities of five state-of-the-art LLMs, including proprietary, open-source, and reasoning models. We probe LLMs in sequential naming and production tasks involving lists of letters and words, adopting a variety of prompting instructions to explore the role of chain-of-thought in the spontaneous emerging of counting strategies. We also evaluate open-source models with the same architecture but increasing size to see whether the mastering of counting principles follows scaling laws, and we analyze the embedding dynamics during sequential enumeration to investigate the emergent encoding of numerosity. We find that some LLMs are indeed capable of deploying counting procedures when explicitly prompted to do so, but none of them spontaneously engage in counting when simply asked to enumerate the number of items in a sequence. Our results suggest that, despite their impressive emergent abilities, LLMs cannot yet robustly and systematically deploy counting procedures, highlighting a persistent gap between neural and symbolic approaches to compositional generalization.

## 1 INTRODUCTION

Many animal species possess an approximate sense of quantity, which allows to coarsely estimate the number of items in a set (Dehaene, 2011). This "number sense" can support the rapid and parallel estimation of the number of objects in visual scenes (Burr & Ross, 2008; Kaufman et al., 1949), but it also allows to estimate the number of events in sequences of flashes and sounds (Dolfi et al., 2024; Philippi et al., 2008) or motor actions (Platt & Johnson, 1971; Cicchini et al., 2016) through serial accumulation mechanisms (Whalen et al., 1999; Gallistel & Gelman, 2000).

In addition to these approximate enumeration abilities, humans also have the unique capacity for exact enumeration, which requires systematically deploying counting procedures (Feigenson et al., 2004). Sequential counting involves a bi-directional mapping between discrete quantities and number symbols that must conform to specific rules: i) one-to-one correspondence between items and symbols, ii) stable order, and iii) cardinality principle (Gallistel & Gelman, 1992). Bi-directionality of the mapping is implied by the deployment of counting both to determine the number of items in a set (*how-many* task) and to create a set of N items (*give-N* task). Crucially, learning to count is considered a key milestone in the development of numeracy, because it implies the understanding of an abstract "successor function" (Sarnecka & Carey, 2008), and it is consistently linked to arithmetic performance and math skills (Koponen et al., 2019). Not surprisingly, it takes several years for children to fully master sequential enumeration, especially in give-N tasks (Sarnecka & Carey, 2008).

Interestingly, despite the remarkable achievements in disparate math domains such as automated theorem proving (Polu & Sutskever, 2020) and discovery of new mathematical conjectures (Davies

et al., 2021), it is still unclear whether modern AI systems possess a true understanding of quantities and numbers (Testolin, 2024). Indeed, it has been repeatedly shown that even the most advanced deep learning models still make striking errors when it comes to basic tasks that require enumerating the number of items in a visual scene (Kajić et al., 2024; Testolin et al., 2025) or the number of letters and words in a sentence (Fu et al., 2024; Xu & Ma, 2024).

In this work, we therefore systematically study the sequential enumeration abilities of state-of-the-art Large Language Models (LLMs), by probing them with both naming (how-many) tasks and production (give-N) tasks involving sequences of discrete elements. We explore different prompting strategies to investigate whether more advanced models can spontaneously deploy counting procedures and whether the responses are still approximately close to the target when explicit counting is forbidden. We also analyze how model size relates to task accuracy to see whether the emergence of more accurate counting skills is directly related to the number of tunable parameters. Finally, in order to gain insights into the inner encoding of counting-related numerical information, we inspect the temporal dynamics of the latent space by analyzing the state space trajectory of embeddings during enumeration, and we also study neuronal tuning functions by measuring how the activation of individual hidden neurons is modulated by changes in the properties of the sequence being counted.

The key contributions of our work are the following: i) We conduct a systematic study of the sequential enumeration capabilities of state-of-the-art LLMs, including both proprietary and open-source models; ii) Besides the most commonly used naming task, which requires to establish the number of items in a given input string, we also probe the models using an original production task, which requires to generate a sequence containing a target number of items; this distinction is particularly relevant considering that naming and production tasks provide different insights into enumeration capabilities (Connor et al., 2024); iii) We implement a variety of prompting strategies to establish whether LLMs can spontaneously deploy counting procedures, and whether explicitly asking to count could result in performance gains compared to the use of more generic instructions; iv) We study whether performance depends on the nature of the items being enumerated, using both uniform and heterogeneous sequences of letters or words; v) For open-source Llama models, we analyze the dynamics of hidden neurons during the generation of item sequences to study the emergence of serial accumulation mechanisms.

## 2 RELATED WORK

Serial counting has long been considered a key benchmark for investigating the computational capabilities of recurrent neural networks. Seminal work in the 1990s showed that recurrent networks trained in token prediction tasks can learn to process simple deterministic context-free languages that require to count the number of elements in a string (Rodriguez et al., 1999), while long-short-term memory (LSTM) networks can exhibit even more robust accuracy and generate sequences of precisely timed spikes (Gers & Schmidhuber, 2000). However, although LSTMs in some cases can generalize to higher numbers than those seen in the training set (Weiss et al., 2018), recent work still suggests that recurrent architectures struggle to learn counting algorithms (El-Naggar et al., 2023). Others have shown that memory-augmented neural networks trained using reinforcement learning can partially extrapolate on give-N tasks, but only up to small numbers (n=15) (Dulberg et al., 2021).

Computational modeling studies have mostly focused on studying parallel visual estimation of numerosity (Chen et al., 2018; Testolin et al., 2020; Creatore et al., 2021), also considering the acquisition of explicit counting skills (Sabathiel et al., 2020) or the role of sequential eye movements (Thompson et al., 2024). Other recent work has shown that multi-modal AI systems have poor parallel visual enumeration capabilities (Rane et al., 2024; Testolin et al., 2025): this phenomenon might be analyzed through the lenses of the binding problem, which arises when processing multiple objects simultaneously using a shared set of representational resources (Campbell et al., 2024).

However, only a few studies investigated sequential enumeration in LLMs. Although it is well known that LLMs often exhibit nontrivial emergent abilities, including mathematical reasoning (Frieder et al., 2023), they appear to have poor counting skills (Delétang et al., 2022), which could partially explain their limitations in tasks that require length generalization (Anil et al., 2022). For example, it has been recently shown that LLMs cannot reliably count the number of letters in a word (Fu et al., 2024) or the number of words in a sentence (Xu & Ma, 2024). Some authors have proposed that these issues could be due to the non-recurrent nature of transformers (Chang & Bisk, 2024), while

others argue that layer normalization and scaling of the attention weights are the two main operations that make it impossible for standard transformers to learn counting in a generalizable way (Ouellette et al., 2023). It has also been proposed that counting in transformers is influenced by vocabulary size or number of unique tokens in the sequence (Yehudai et al., 2024), though the latter was confounded with sequence length in the simulations. Another recent work has shown that counting in LLMs is more challenging when the correct output is a low-probability piece of text (e.g., rarely used numbers) (McCoy et al., 2023). A concurrent line of research has focused on the computational advantages provided by chain-of-thought (CoT) prompting methods (Wei et al., 2022b). Indeed, although LLMs are based on the transformer architecture and therefore inherit its computational limitations (Liu et al., 2022), CoT dramatically improves their accuracy in tasks that are hard for parallel computation thanks to the iterative processing of intermediate reasoning steps (Li et al., 2024).

## 3 EXPERIMENTAL SETUP

### 3.1 MODELS TESTED

We consider several LLMs spanning different sizes and architectures. For proprietary models, we include the newest GPT5 reasoning model [gpt-5-2025-08-07] and the GPT4 model [gpt-4.1-2025-04-14] developed by OpenAI (OpenAI, 2024), and the most recent Gemini family model [gemini-2.5-pro-preview-03-25] developed by Google (Team et al., 2023). For open-source models, we include three variants of increasing size from the Llama3 family released by Meta (Dubey et al., 2024): [Llama-3.2-3B-Instruct], [Llama-3.1-8B-Instruct] and [Llama-3.3-70B-Instruct], as well as the recent Qwen reasoning model [QwQ-32B] developed by Alibaba (Yang et al., 2024). Information about the computing hardware used in the experiments is provided in Supplementary Section B.

### 3.2 SEQUENTIAL ENUMERATION TASKS

We examine two distinct types of sequential enumeration tasks that cognitive scientists have proposed to probe basic numerical skills: numerosity naming (Revkin et al., 2008) and numerosity production (Whalen et al., 1999; Sella et al., 2016). These tasks allow for assessing the representation and manipulation of numerical quantities, and are adapted here to evaluate the extent to which AI models can operate over numerical sequences of discrete elements. In the naming task, the model is presented with a string representing a sequence of elements and is required to output the number of elements contained in the sequence. The numerosity production task reverses this process: the model is given a target numerical value and is asked to generate a sequence containing the specified number of elements. This tests the model's ability to translate abstract numerical representations into concrete, countable outputs. Previous work (Dulberg et al., 2021) studied a production task using small numbers; here we test models on target numbers ranging from 10 to 100 in increments of 10 to ensure systematic coverage across enumeration ranges.

We explore enumeration skills on two types of discrete elements: letters and words. For letters, stimuli are randomly sampled from the standard English alphabet characters. For words, stimuli are randomly selected from a curated list of five-letter English words to ensure that all elements have the same length and avoid confounds due to variability in non-numerical magnitudes (Testolin et al., 2020). Importantly, all words were chosen to ensure a one-token-per-word correspondence to guarantee that the model's task emphasizes sequential enumeration rather than language modeling or token segmentation (Zhang et al., 2024). In both conditions, we explored sequences containing both homogeneous and heterogeneous elements. Further details about the creation of testing sequences can be found in the Supplementary Section C.

### 3.3 PROMPTING STRATEGIES

We designed a structured set of system prompts made up of four key sections: 1) *contextual background*, 2) *task goal*, 3) *counting strategy*, and 4) *response formatting*.

**Contextual Background.** Each prompt begins by placing the model in the role of a participant in a psychometric experiment through the sentence "You are the subject of a psychometric experiment designed to investigate enumeration capabilities. Do your best to be accurate.". This narrative framing

is intended to engage the model in a cooperative, goal-directed task where accuracy and compliance are expected.

**Task Goal.** This section defines the enumeration objective. For the naming task, the model is asked to count the number of repetitions in a given sequence (″`I will ask you how many letters/words are there in a string.`″) An example of the user message paired with this condition: "How many 'table' are there in '`table table table table table table table table table`'?". For the production task, the model is prompted to generate a target number of repeated letters or words (″`I will ask you to write some letters/words a given number of times.`″). An example of the user message paired with this condition: "Write the letter `A` 10 times". This setup allows us to examine both forward (sequence-to-number) and reverse (number-to-sequence) mappings. Additional prompt examples are provided in Section D.2. As a control condition, we also tested whether the most powerful LLMs could count via coding (details can be found in Supplementary Section G.5).

**Counting Strategy.** The key manipulation across prompt types lies in the strategy the model is instructed—or allowed—to use for enumeration. We distinguish four conditions:

- **Explicit Counting:** The model is obligated to count carefully and to use any supportive strategy (e.g., markers or numerical annotations). *Instructional cue: "You should generate markers, numbers or any other symbols that may help you keep track."* This tests the model's capacity for deliberate and systematic enumeration.

- **Spontaneous Counting:** The model receives no guidance regarding counting and must decide autonomously whether and how to count. *Instructional cue: None (no explicit instruction about counting is provided).* This condition assesses whether systematic counting behavior can emerge in the absence of explicit instruction.

- **Mental Counting:** The model is allowed to count, but only "mentally", without producing any visible symbols to support counting. *Instructional cue: "You are allowed to count in your mind but please do not count explicitly: you cannot generate markers, numbers or any other symbols that may help you keep track."* This probes the model's capacity for covert enumeration and internal working memory usage.

- **Forbid Counting:** The model is explicitly instructed not to count or use any form of structure that aids counting. *Instructional cue: "You are not allowed to count and you cannot generate markers, numbers or any other symbols that may help you keep track."* This probes the approximate enumeration abilities of the model, in analogy with the articulatory suppression paradigms used in cognitive science to study temporal estimation when counting is precluded (Rattat & Droit-Volet, 2012).

**Response Formatting.** At the end of each system prompt, we ask the models to mark their responses within the special tags "<my_answer" and "</my_answer>" to facilitate the post-processing of the output. For the smaller Llama models (Llama3 3B and Llama3 8B), response formatting is tailored to each model so that the model can perform the best alignment with the instructions. More details can be found in Supplementary Section D.1.

### 3.4 PERFORMANCE EVALUATION

Model responses were parsed semi-automatically using Python scripts (details in Supplementary Section F). Two metrics then quantified task performance. Accuracy was defined as the proportion of trials where the parsed output exactly matched the target numerosity; invalid trials were counted as incorrect. Mean Absolute Error (MAE) was instead computed as: $\text{MAE} = \frac{1}{n} \sum_{i=1}^{n} |C_i - \hat{C}_i|$ where $c$ is the target numerosity and $\hat{c}$ is the model's reported count. To prevent bias toward low-count performance, errors exceeding 512 were capped at 512. Further details, including per-condition token limits and invalid trial counts, are provided in Supplementary Section E.

### 3.5 LATENT SPACE DYNAMICS AND NEURONAL TUNING PROFILES DURING ENUMERATION

To investigate how the Llama70B model internally tracks the counted items we analyzed the dynamics of its hidden states during token generation across the four prompting strategies (inference details in

Supplementary Section E). Specifically, we aimed to identify whether an internal counter mechanism emerges in the model, and to determine under which conditions such a structure is most apparent. We therefore applied principal component analysis (PCA) to the last-layer embeddings extracted during the stepwise generation process. By reducing the high-dimensional activation space to a lower-dimensional trajectory, PCA enables us to examine structured temporal dynamics and assess whether consistent patterns, such as linear progression or cyclic structure, are present across counting steps. This approach is widely used in neuroscience to analyze neuronal population activity, and it has proven effective in uncovering latent dynamical structures underlying behaviorally relevant computations (Cunningham & Yu, 2014). Separate PCA models were fit for each condition to preserve potential condition-specific representational structure. To analyze temporal dynamics, we extracted PC's time series for each target numerosity (for trials with sequence length exceeding the target numerostiy, we clipped the time series at the target number). While the first two PCs are presented for clear 2D visualization and often capture the majority of variance related to behaviorally relevant computations, we also inspected higher-order components (see Supplementary Section G.6).

## 4 RESULTS

### 4.1 TASK PERFORMANCE

Figure 1 shows the enumeration accuracy across all prompting conditions for naming and production tasks involving homogeneous sequences of elements[1] Proprietary models consistently achieve the highest accuracy, though open-source models also excel in production tasks when explicitly asked to count. However, accuracies significantly drop in naming tasks even when the models are explicitly asked to count, suggesting that it is more challenging to enumerate a given list of items rather than producing one from scratch. Accuracy also generally declines under all other prompting conditions, showing that none of the models can reliably enumerate the elements of a sequence when not explicitly instructed to count. However, the MAE plots show that when proprietary models make errors, they still get reasonably close to the target number.

Notably, in the spontaneous counting condition, none of the models exhibits systematic counting behavior (see Supplementary Section G.6). When the models do not systematically count, the gap between the naming and production tasks is reduced, suggesting that in such conditions the LLMs rely on similar approximate estimation mechanisms. Nevertheless, it is interesting to note that GPT and Gemini can still provide answers that are close to the target, as indexed by their relatively low MAE. It is also interesting to note that overall the performance for the mental counting condition is slightly higher than that observed in the spontaneous and forbid counting conditions, suggesting that the models can somehow improve their internal encoding of numerosity when prompted to do so. To better investigate how different strategies impact behavioral performance, we computed a scale-invariant accuracy measure (Normalized Absolute Error) for each counting condition and conducted a Friedman non-parametric test, followed by Bonferroni-corrected pairwise comparisons (details can be found in Supplementary Section G.3). In general, we can also see that LLMs are more accurate in naming and production tasks involving words rather than letters. This result is aligned with recent findings that highlight the importance of tokenization in counting precision (Zhang et al., 2024). Indeed, our selected set of words has a one-to-one correspondence with the tokens provided as input to the models, while sequences of letters can be tokenized differently (e.g., grouping a set of letters together) depending on the model specifications.

For the Llama family we also studied the relationship between model size and counting performance, to investigate whether counting skills can be characterized as an emerging ability (Wei et al., 2022a). As model size increased from 3 to 70 billion, accuracy improved substantially, increasing from approximately 0.10 to 0.24, and MAE similarly decreased from over 200 in the smallest model to around 70 in the largest (see Supplementary Section G.2). However, this pattern shows that counting is not strictly emergent in the classic sense of a sharp threshold or phase transition, but rather gradually develops as model capacity increases (Schaeffer et al., 2023). Finally, one should note that the Qwen performance is still remarkable compared to that of other models, considering that it has a much smaller number of parameters.

---

[1]Results on heterogeneous sequences are reported in Supplementary Section G.4. Performance is qualitatively similar, though homogeneous sequences seem slightly more challenging to enumerate (especially for the naming task), probably because in uniform sets individual items are less distinguishable.

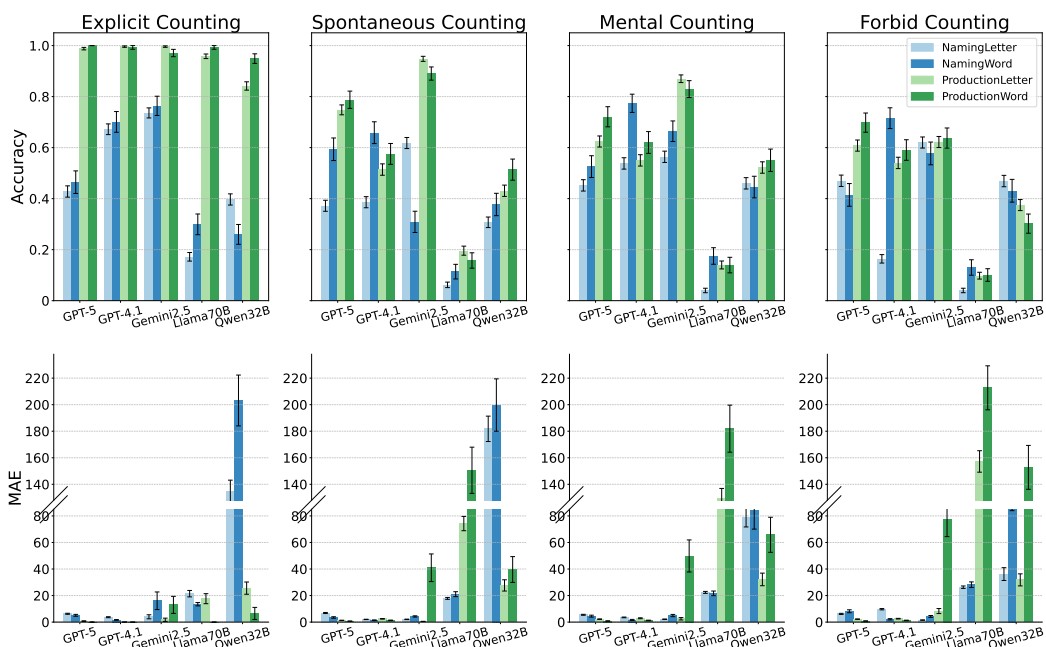

Figure 1: Comparison of accuracy and mean absolute error (MAE) across different models and prompting conditions. Bars represent accuracy or MAE for each model (x-axis) across the four tasks: naming/production × letter/word. Error bars indicate the variability of the estimates: for accuracy, binomial standard errors are shown; for MAE, standard errors of the mean are shown.

## 4.2 ANALYSIS OF ENUMERATION ERRORS

For all models, the variance of the responses shows a consistent trend that aligns with accuracy across different prompting conditions, with errors increasing as the target numerosity increases (see Figure 2). Response errors for GPT and Gemini models are similar and generally close to the target number, even though in the forbid counting condition Gemini tends to overestimate the number of letters generated in the production task, while GPT4.1 tends to underestimate (see Supplementary Figure 5). Llama systematically underestimates in the naming task and overestimates in the production task, while Qwen does not exhibit specific biases. A Friedman test was carried out on the absolute errors as a function of condition, followed by Bonferroni-corrected pairwise comparisons, showing that counting strategies indeed significantly affect the amount of errors (see Supplementary Section G.3).

## 4.3 NEURAL DYNAMICS DURING SEQUENTIAL ENUMERATION

The PCA analysis revealed that a small number of components can account for a substantial proportion of the variance in model activations across all counting conditions: PC1 alone explained 17% (explicit) and 47% (mental) of the total variance. Figure 3 shows how the first and second Principal Components (PC1 and PC2) change across counting steps for the explicit and mental counting conditions (the other conditions are shown in Supplementary Fig. 8). In the mental counting condition, PC1 and PC2 show a strong correlation with the generated steps (computed for trials with the same target number and then averaged across targets), with Spearman correlations exceeding 0.83 and - 0.88, respectively (both $p < 0.001$), consistent with a step-tracking signal that may reflect an emergent internal counter. Interestingly, the explicit condition strongly diverges from mental counting even along the first component: the PC1 does not show (piece-wise) monotonicity as a function of step number but is characterized by a periodic trend with large dips at steps that are multiples of ten. The latter finding might be linked to the observation that LLMs use Fourier features to compute basic arithmetic operations (Zhou et al., 2024) and/or to the presence of biases for decade numbers in training corpora (Testolin et al., 2025). To explore this phenomenon we conducted a control experiment where the model was prompted to count in base-16 with different target numerosities (either multiples of 10, or multiples of 16; see details in Supplementary Section G.6.1). The results

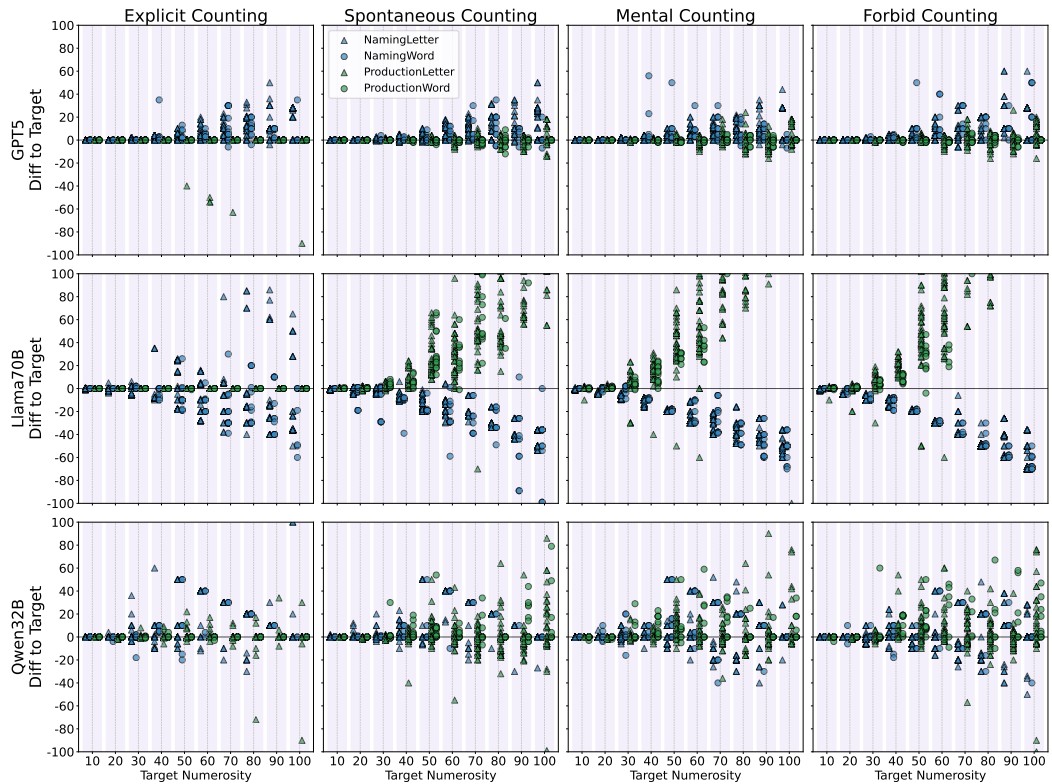

Figure 2: Scatter plots of enumeration errors (response – target value) across different prompting conditions for a selected subset of models. Each panel shows the response differences plotted against the target values, with distinct markers of different colors representing different task-stimulus combinations: blue for the naming task and green for the production task; triangles for letters and filled circles for words.

suggest that the observed periodicity is partially due to the decimal-counting instruction at inference time, but also reflects a statistical bias in the training corpora, where base-10 number words (e.g. "twenty," "thirty") are far more frequent than other numerical constructions.

Despite high behavioral accuracy in the explicit condition, PCA revealed little systematic accumulation structure in the leading components. PC1 explained less variance, and PC1 seems to mainly account for tracking the decade number changes during generation. The internal dynamics underlying the mental counting condition instead appear more structured, especially when looking at the geometry of the state-space defined by the first two PCs (panel B in Figure 3): we can observe a distinctive non-linear trajectory (horseshoe pattern), suggesting that the internal representation of counting evolves through several distinct phases. Nevertheless, the smooth overall trajectory suggests a consistent underlying computational mechanism driving the enumeration, and the clear structure visible in just two dimensions indicates that these principal components capture meaningful aspects of the internal counting dynamics. Moreover, the representational geometry has a striking resemblance to the ordering of numerosities along a curved manifold recently observed in human neuroimaging (Karami et al., 2025). Note also that the more compact range of PC values in mental counting suggests a more efficient (i.e., lower-dimensional) representation of quantity compared to explicit counting, suggesting that the model employs fundamentally different computational strategies for explicit versus mental counting. Overall, these findings suggest that structured, low-dimensional neural trajectories resembling internal counters emerge most clearly in the mental counting condition, despite the absence of explicit instructions, while explicit prompting leads to high performance via less interpretable, potentially token-based mechanisms.

To identify neurons sensitive to sequence progression, we further analyzed hidden neuron activations across the four experimental conditions. For each trial, we selected responses with at least 100

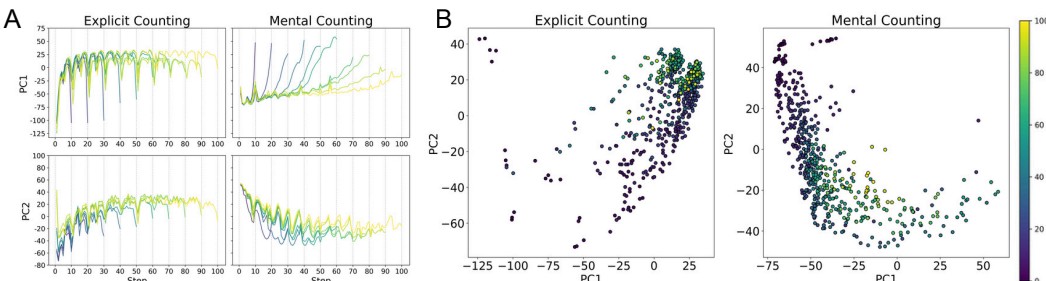

Figure 3: Panel A: Trajectories of the first two principal components computed over hidden states, separately for explicit counting and mental counting conditions. Each subplot shows the trajectories along PC1 and PC2 as a function of generation step, with colors indicating the target numerosity. Panel B: 2D presentation of the PCA results. The marker's color gradient reflects generation order across generation steps.

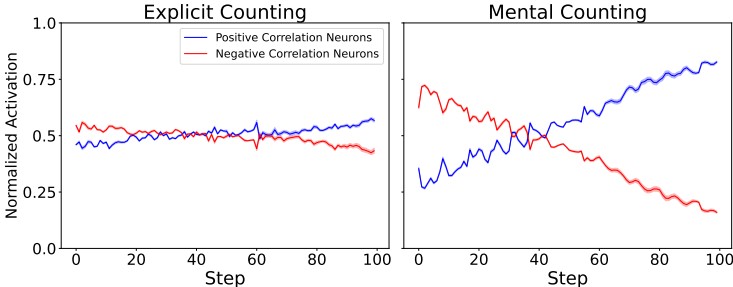

Figure 4: Population dynamics of unit activations across stepwise counting. For each condition, we plotted the average activation of selected neurons within steps made from 1 to 100. The activation was normalized and shaded with standard error for better visualization.

generated steps and truncated them to the first 100 steps. Each neuron's activation over generations was extracted and reshaped across trials to compute Pearson correlations with step indices. We identified the 500 neurons with the highest positive and 500 with the most negative correlations.

As shown in Figure 4, we found that the activations change with the steps in two opposite directions, as expected in the mental counting condition. In the explicit condition this trend is less marked, with both groups of neurons' activation level oscillating around the mean level. This suggests that in the explicit counting condition, there is not a population of neurons encoding the number of steps generated via activation. On the other hand, mental counting engages specialized populations of neurons with opposite activation trajectories. The variance for explicit counting is almost zero, while for mental counting, the variance increases after generation step 40.

## 5 DISCUSSION AND CONCLUSIONS

Overall, the variation in task performance across different prompting strategies highlights the extent to which LLMs rely on explicit cues and structured prompting to succeed in numerosity-related tasks. Under the explicit counting condition, all models performed considerably better, with top-tier proprietary models achieving near-ceiling accuracy and minimal MAE, but only in the production task. This suggests that providing clear and structured prompts allows the models to exhibit a more consistent enumeration behavior, but also shows that even the most advanced models still struggle with counting the number of elements in a given list.

The significant drop in performance observed in the spontaneous counting condition further shows that while some models may possess an emergent capacity to self-initiate counting, such behavior is extremely rare and inconsistent across tasks, revealing a key limitation of LLMs in the ability

to spontaneously enumerate sequences of elements. Finally, the slightly higher performance in the mental counting condition compared to the spontaneous and forbid counting conditions further highlights the importance of prompt framing: in line with previous findings, most models appear to be sensitive not just to the operational structure of the task but also to the cognitive stance implied by the prompt (Battle & Gollapudi, 2024).

Our state space analysis provides empirical support for this assertion, revealing fundamentally different computational strategies: only the mental counting condition showed a smooth, continuous trajectory resembling an accumulation process, suggesting a more analog representation of quantity similar to serial accumulation mechanisms implemented in biological circuits (Gallistel & Gelman, 2000; Nieder et al., 2006). Notably, this gradual accumulation disappears during explicit counting, suggesting that in such condition the LLM rather exploits an associative chaining mechanism that allows it to keep track of the items by only relying on the previously generated symbolic number, thereby exploiting a surface-level token prediction strategy rather than maintaining an internal counter. Given the suboptimal performance of Llama models, it is also evident that these LLMs lack a mechanism to compare the accumulation signal to the target symbolic number that is memorized during prompting instruction: this issue can be put in relation to the acquisition of counting skills during human development, where children often simply recall the counting sequence by rote memory, without having grasped the number semantics, and thus often cannot properly terminate the counting procedure because they lack a bi-directional mapping between numerosity representations and symbolic numbers (Sarnecka & Carey, 2008).

Nevertheless, we acknowledge that our analysis of the neural dynamics represents only a first step to get a mechanistic explanation of sequential counting processes: while we identified behaviorally-relevant neural activity patterns, future work should employ more sophisticated circuit-level analyses, such as those based on causal abstraction (Geiger et al., 2024). In this respect, we should also note that the use of proprietary models in scientific research poses challenges Palmer et al. (2024), since we do not have full access to the internal functioning of these systems (e.g., models like GPT or Gemini could in principle exploit external tools based on symbolic counting algorithms to carry out enumeration tasks).

Our study also offers a nuanced perspective on the recent theoretical work by Yehudai et al. (2024). Indeed, their analysis proposes that transformers can reliably count only if the embedding dimension $d$ is larger than the vocabulary size $m$. Our scenario presents a key difference: while their study confounds vocabulary size with sequence length, our sequences use a single repeated token, isolating sequence length as the primary factor in counting errors. This supports and clarifies their broader theoretical point about the challenges of long-range dependencies. Interestingly, our results also indicate that model size is a less critical factor than architectural or training differences, as smaller models sometimes outperformed larger ones. This finding challenges the specific confound in the empirical demonstration proposed by Yehudai et al. (2024), at the same time extending their theoretical framework by highlighting sequence length as a dominant variable in real-world LLMs.

In conclusion, our work shows that while in some circumstances LLMs develop internal activation dynamics resembling serial accumulation mechanisms, these signals are not precise enough to guarantee accurate enumeration performance. Alternatively, when the LLMs rely on explicit counting, they might in fact deploy surface-level token prediction capabilities that might prevent a deeper understanding of the task semantics. It is conceivable that proficient mastery of counting and a deeper understanding of the properties of the number system will require grounding of number concepts into sensorimotor experience, as claimed by mainstream theories in cognitive science (Lakoff & Núñez, 2000).

## 6 ETHICS STATEMENT

The authors are well aware of the code of ethics and have complied with it.

## 7 REPRODUCIBILITY STATEMENT

The authors ensured the statistical power and reproducibility of the study by running the experiments and analyses under various task conditions, on a sufficiently high number of trials. The code for reproducing the results is provided in the supplementary information. The authors will publicize the code and data in the supplementary information upon acceptance.

## 8 THE USE OF LARGE LANGUAGE MODELS

The authors acknowledge the use of LLMs in polishing writing.

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

## A MATERIALS AND SOURCE CODE

Models' responses are stored in the pickle file of each model, and embedding trajectories can be found in our anonymized drive due to size limit. Scripts for collecting the results can also be found there.

## B COMPUTATIONAL HARDWARE AND CLOUD APIs

Data collection for open-source models was conducted using a computing cluster equipped with seven NVIDIA L40s GPUs, as well as Google Cloud instances featuring A100 GPUs. For proprietary models, responses were obtained through official APIs provided by OpenAI, while Gemini model outputs were collected using the Vertex AI platform.

## C CREATION OF TESTING SEQUENCES

### C.1 WORD SELECTION

We first selected a set of words that are 5 letters long, and we created both homogeneous and heterogeneous sequences of words by randomly sampling from this list. A critical prerequisite for a valid analysis of counting behavior and its underlying neural dynamics is to ensure that the linguistic units of counting are represented as single tokens within the model's vocabulary. To this end, we verified the tokenizer for each model in our study to confirm that all target words (e.g. `'table'` in the previous example) were encoded as single tokens. This control ensures that the observed neural dynamics reflect the cognitive process of counting and are not artifacts of sub-word tokenization or compositional encoding.

### C.2 HOMOGENEOUS STIMULI

The homogeneous stimuli were created by randomly sampling a word from the curated list or a random letter, then repeating it the target number of times for the naming task. In the production task, the model was asked to write the sampled word or letter the target number of times.

### C.3 HETEROGENEOUS STIMULI

The non-uniform target string in this case was formed by different letters or words. For the naming task, both sequences of letters and words are tested, while for the production task, only heterogeneous letters were asked [2]. Rather than sampling one word and repeating it the target number of times, we sampled the target number of different words and formed a string. While for the letters, repetition of the same letter is allowed. Results with heterogeneous stimuli can be found in Section G.4.

## D PROMPTING DETAILS

### D.1 RESPONSE FORMATTING

As discussed in Section 3.3, our system message consists of four components: (1) *contextual background*, (2) *task goal*, (3) *counting strategy*, and (4) *response formatting*. To maximize the performance of each LLM and ensure that responses adhere to the desired format, we specifically investigated the design of the *response formatting* component for models that exhibited lower proportions of valid outputs. These models include Llama-3B and Llama-8B, which underperformed compared to others in terms of valid trial rates. For reference, the proportions of valid trials achieved by the higher-performing models are as follows: Gemini 2.5 Pro achieved 98.3%, GPT-5 and GPT-4.1 achieved 100.0%, Llama-70B achieved 93.3% and Qwen-32B achieved 87.5%.

---

[2] Heterogeneous word generation poses a significant challenge to our automatic response parsing pipeline. Even with human inspection, tricky cases still exist. For instance, "OK, I will start generating 20 random words: apple, banana, cat, ice cream....". On the other hand, for letters, we explicitly asked the model to generate a space before each letter, which can be parsed easily with our pipeline.

To optimize the response formatting instructions, we explored several strategies: (1) varying the position of the response formatting (either preceding or following the task goal); (2) experimenting with alternative phrasings of the instruction; and (3) providing an explicit example of a correctly formatted response. We conducted a controlled experiment to evaluate the effect of each variation. For both the naming and production tasks, we executed 10 trials per numerosity. Each set of 10 trials consisted of 5 trials using words and 5 using letters, both sampled randomly. The response formatting instruction that yielded the highest proportion of valid outputs was selected.

The alternative phrasings we explored are:

1. Format/write/give your response between an opening tag <my_answer> and a close tag </my_answer>.

2. Please format all your answers by wrapping them in a <my_answer> opening tag and a </my_answer> closing tag.

3. Begin/start answering with <my_answer> and finish/stop with </my_answer>.

For Llama-3B, the best response formatting was "Format your response between an opening tag <my_answer> and a close tag </my_answer>" without changing position and giving explicit examples. For Llama-8B, "Please format all your answers by wrapping them in a <my_answer> opening tag and a </my_answer> closing tag. For example, if the answer is '76', your response should be <my_answer>76</my_answer>." resulted in giving more valid trials. After applying those response formatting, Llama-3B has 75.8% of valid trials and Llama-8B has 83.5% of valid trials.

### D.2 PROMPT EXAMPLES

We provide a more complete set of per-trial prompts in the shared drive.

## E INFERENCE DETAILS

### E.1 TOKEN LIMITS

To accelerate the data collection process and minimize computational overhead, we imposed upper limits on the number of tokens generated by the models. Specifically, for the naming task, the maximum number of tokens allowed was set to 512 for the word condition and 256 for the letter condition. For the production task, the corresponding limits were 1024 tokens for the word condition and 512 tokens for the letter condition. Those limits are further adjusted during generation (details in Section F). For Qwen-32B, those token limits apply to the tokens after the '</think>' token, and we set a 2048 token limit for the reasoning process to prevent a non-stopping think loop.

### E.2 TOKENIZATION CONTROL

To examine internal model representations, we extracted the hidden states from the final layer of the Llama-70B model, which consists of 8192 neurons per token. These activations provide a high-dimensional embedding of the model's internal processing during generation.

To isolate representational patterns independent of behavioral outcomes, we conducted a separate experiment from the main analyses presented in Section 4.1 and 4.2. In this auxiliary run, we selected a single concrete word, "apple," and focused exclusively on the production task. For each numerosity level, we generated at least 10 valid trials. This approach allowed for controlled analysis by eliminating the variability introduced by the tokenization of letter sequences. Unlike words, letter sequences (e.g., "M," "MM," and "MMMM" are all represented by a single token) are subject to inconsistent tokenization. By using a single, concrete word, we ensured stable token boundaries and reliable extraction of hidden states.

## F AUTOMATIC RESPONSE PARSING PIPELINE

Each trial was checked for the presence of the special tags <my_answer> and </my_answer>. If missing, the trial was marked *invalid*, and the model was allowed up to nine additional attempts

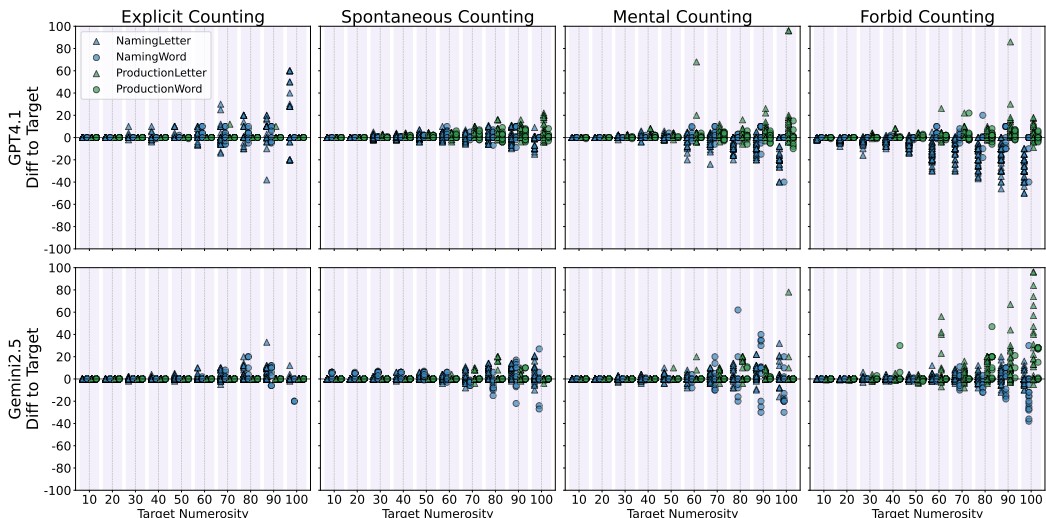

Figure 5: Scatter plots of enumeration errors (response – target value) across different prompting conditions for a selected subset of models. Each panel shows the response differences plotted against the target values, with distinct markers of different colors representing different task-stimulus combinations: blue for the naming task and green for the production task; triangles for letters and filled circles for words.

(ten total) to produce a valid response. In the production task, to prevent non-terminating output (e.g., repetition), we enforced task-specific token limits. If the model reached this limit, the token budget was halved and retried until it dropped to 512 tokens. Hitting the 512-token ceiling on three consecutive attempts rendered the trial invalid; otherwise, up to ten retries were allowed.

Post-generation, valid responses were parsed based on task type. For the naming task, we extracted content between the answer tags and checked if it contained a single integer. Non-integer outputs were flagged for manual review. In the production task, we used regular expressions to extract and count only target tokens (letters or words) within the tags; extraneous text also triggered manual inspection. We made sure that spelling errors in the generated words were not regarded as wrong responses by implementing a post-processing pipeline measuring edit (Levenshtein) distance to find cases where the output word is only moderately different from the target word, and making sure that these were counted as correct trials. The results showed that all words generated have 0 distance from the target word, showcasing no spelling errors at all.

# G RESULTS

## G.1 BEHAVIOR PATTERNS

### G.1.1 SYSTEMATIC COUNTING

We investigated whether the models engaged in systematic counting when required (explicit counting) and when contextually beneficial (spontaneous counting). This analysis focused on the production task. The naming task was excluded, as the models frequently returned errors, potentially due to either repeating the target string incorrectly or failing to count. The current study design does not allow us to disentangle these factors.

In the explicit counting condition, all models systematically counted on every valid trial. GPT-5 and Gemini 2.5 produced error-free responses in all valid cases. GPT-4.1 made 3 errors, Llama70B made 1 error, and Qwen-32B made 57 errors. Of GPT-4.1's errors, two were due to grouping instead of systematic counting. Both GPT-4.1 and Llama-70B made an error on the word *under*, despite counting systematically; in both cases, they under-enumerated, suggesting that semantic information may interfere with counting accuracy. Qwen-32B's errors arose not from a failure to count but from mistakes during the copying of letters and words from the think loop to our special tags without

systematic counting again inside the special answer tags. In the spontaneous counting condition, among all the results, only Gemini 2.5 Pro deployed systematic counting in the production task for one trial (ID. 5385).

### G.1.2 EXAMPLES OF COUNTING BEHAVIOR

We provide some representative examples of models' counting behaviors. Full responses can be found in the pickle files we provide.

1. **Grouping** — *Gemini 2.5 Pro Production ID:197*
   "Okay, I will write the letter J exactly 70 times, using numbered groups to keep track.

   1 J J J J J J J J J J (10)
   2 J J J J J J J J J J (20)
   3 J J J J J J J J J J (30)
   4 J J J J J J J J J J (40)
   5 J J J J J J J J J J (50)
   6 J J J J J J J J J J (60)
   7 J J J J J J J J J J (70)

   I have written the letter J exactly 70 times."

2. **Systematic Counting** — *Gemini 2.5 Pro Production ID:5*
   "1. Q
   2. Q
   3. Q
   4. Q
   5. Q
   6. Q
   7. Q
   8. Q
   9. Q
   10. Q"

3. **Grouping and Systematic Counting** — *Gemini 2.5 Pro Production ID:581*
   "Okay, I will write the letter U exactly 90 times. I will use numbering in groups of 10 to keep track.

   1 2 3 4 5 6 7 8 9 10
   U U U U U U U U U U (10)

   11 12 13 14 15 16 17 18 19 20
   U U U U U U U U U U (20)

   21 22 23 24 25 26 27 28 29 30
   U U U U U U U U U U (30)

   31 32 33 34 35 36 37 38 39 40
   U U U U U U U U U U (40)

   41 42 43 44 45 46 47 48 49 50
   U U U U U U U U U U (50)

   51 52 53 54 55 56 57 58 59 60
   U U U U U U U U U U (60)

   61 62 63 64 65 66 67 68 69 70
   U U U U U U U U U U (70)

   71 72 73 74 75 76 77 78 79 80
   U U U U U U U U U U (80)

   81 82 83 84 85 86 87 88 89 90
   U U U U U U U U U U (90)

   I have written the letter U exactly 90 times."

4. **Wrong Repeating and Correct Counting** — *GPT-4.1 Naming ID:405*
   "Let me repeat the string and use markers to count:
   u u u u u u u u u u u u u u u u u u u u u u u u u u u u u u u u u u u u
   Now, let's count by grouping into 5s for clarity:

1 2 3 4 5 6 7 8 9 10 11 12 13 14 15 16 17 18 19 20 21 22 23 24 25 26 27 28 29 30 31 32
Final count: 32"

5. **Correct Repeating but Wrong Counting** — *GPT-4.1 Naming ID:500*
"Let's repeat the string:
storm storm storm storm storm storm storm storm storm storm
storm storm storm storm storm storm storm storm storm storm
storm storm storm storm storm storm storm storm storm storm
storm storm storm storm storm storm storm storm storm storm
storm storm storm storm storm storm storm storm storm storm
storm storm storm storm storm storm storm storm storm storm
storm storm storm storm storm storm storm storm storm storm

Now, let's count the words using markers (I'll use '|' for every 10 counts):
storm storm storm storm storm storm storm storm storm storm | (10)
storm storm storm storm storm storm storm storm storm storm | (20)
storm storm storm storm storm storm storm storm storm storm | (30)
storm storm storm storm storm storm storm storm storm storm | (40)
storm storm storm storm storm storm storm storm storm storm | (50)
storm storm storm storm storm storm storm storm storm storm | (60)
Each line is 10, and there are 7 lines. Let's check: 1st line: 10 2nd line: 10 3rd line: 10 4th line: 10 5th line: 10 6th line: 10 7th line: 10
10 + 10 + 10 + 10 + 10 + 10 + 10 = 70
Final count: 70"

## G.2    ACCURACY AND MAE WITH MODEL SIZES

We investigated the effect of model scaling on sequential enumeration tasks using models from the Llama family, specifically Llama-3B, Llama-8B, and Llama-70B. We selected the Llama family for this comparison because these models share the same architecture and training objective, differing only in model size, that is, primarily through the number of layers and hidden units.

We evaluated performance using two metrics: accuracy and mean absolute error (MAE). As shown in Figure 6, both accuracy and MAE improve progressively with model size. This indicates that the ability to perform sequential enumeration does not exhibit a sharp emergence but rather improves gradually as model capacity increases within the tested size range. These findings suggest that sequential enumeration is not an emergent ability in the strict sense, but instead scales smoothly with model size in this family.

## G.3    FRIEDMAN TEST

We conducted a comprehensive statistical analysis to compare the performance of four different counting conditions (explicit, spontaneous, mental, and forbid) across five models using Normalized Absolute Error (NAE) as the primary metric. NAE is calculated as the absolute difference between the target count and the model's predicted count, normalized by the target count: NAE = |target - predicted| / target. This metric provides a scale-invariant measure of counting accuracy, where values closer to 0 indicate better performance. We employed the non-parametric Friedman test since the data did not meet normality assumptions. Effect sizes were quantified using Kendall's W for the overall test and rank-biserial correlation for pairwise comparisons. When significant differences were detected, post-hoc pairwise Mann-Whitney U tests with Bonferroni correction ($\alpha = 0.0083$) were conducted to identify specific method differences.

The statistical analysis revealed significant differences in counting performance across methods for all five models, though with varying effect sizes and patterns. Table 1 shows the NAE across different models and counting methods. GPT-5 showed the best performance by being error-free in the explicit word production task. The overall effect size is small (W = 0.02), and Post-hoc pairwise comparisons with a Bonferroni correction revealed that the source of this significance stemmed specifically from the 'explicit' method. GPT-4.1 demonstrated the second-best overall performance with consistently low NAE values (ranging from 0.02 to 0.08) and showed a medium effect size (W = 0.14), with explicit counting significantly outperforming all other methods. Gemini 2.5 Pro exhibited similarly

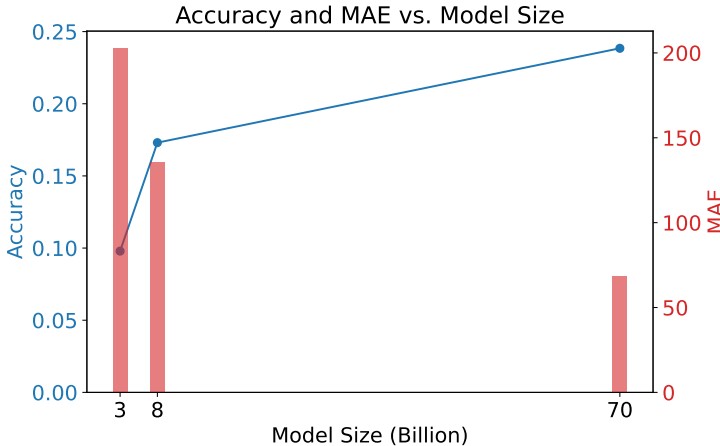

Figure 6: Comparison of model performance across different Llama model sizes. Accuracy is shown as a blue line plot (left y-axis), and Mean Absolute Error (MAE) is represented by red bars (right y-axis). As model size increases from 3B to 70B, accuracy generally improves while MAE decreases, indicating better overall performance with larger models.

Table 1: NAE summary statistics by model and counting method

| Model | Explicit | Spontaneous | Mental | Forbid |
|-------|----------|-------------|--------|--------|
| GPT5 | 0.05 ± 0.10 | 0.05 ± 0.08 | 0.05 ± 0.09 | 0.06 ± 0.10 |
| GPT4.1 | 0.02 ± 0.06 | 0.03 ± 0.04 | 0.04 ± 0.07 | 0.08 ± 0.11 |
| Gemini2.5pro | 0.07 ± 0.66 | 0.08 ± 0.53 | 0.09 ± 0.58 | 0.17 ± 0.90 |
| Llama70b | 0.21 ± 0.71 | 0.69 ± 1.24 | 1.03 ± 1.58 | 1.32 ± 1.91 |
| Qwen32b | 1.38 ± 3.33 | 1.71 ± 3.31 | 0.90 ± 2.31 | 0.79 ± 2.07 |

strong performance (NAE values 0.07-0.17) but with a smaller effect size (W = 0.05), showing that explicit counting was significantly better than spontaneous, mental, and forbid methods. In contrast, Llama 70B displayed the most pronounced method differences with a large effect size (W = 0.37) and substantially higher NAE values, where explicit counting (NAE = 0.21) significantly outperformed all other methods, with performance deteriorating progressively through spontaneous (0.69), mental (1.03), to forbid (1.32). Qwen 32B showed an interesting pattern where mental (0.90) and forbid (0.79) methods performed better than explicit (1.38) and spontaneous (1.71) methods, though the overall effect size remained small (W = 0.03).

In summary, our findings demonstrate that the counting method significantly affects performance across all tested language models, with GPT-4.1 and Gemini 2.5 Pro showing superior accuracy and consistent preference for explicit counting, while Llama 70B exhibited the strongest method sensitivity, and Qwen 32B uniquely benefited from implicit counting approaches.

## G.4 HETEROGENEOUS STIMULI

We also tested how models perform when the stimuli are heterogeneous (*i.e.,* a string of random words or letters). For the production task, only random letters are tested, as it is challenging to find a reliable way to automate the answer validation process. As shown in Fig. 7, similar trends can be observed as when the models were tested with homogeneous stimuli. Notably, GPT-5 delivered perfect responses in the explicit and spontaneous conditions. Qwen was flagged as having an insufficient number of valid trials.

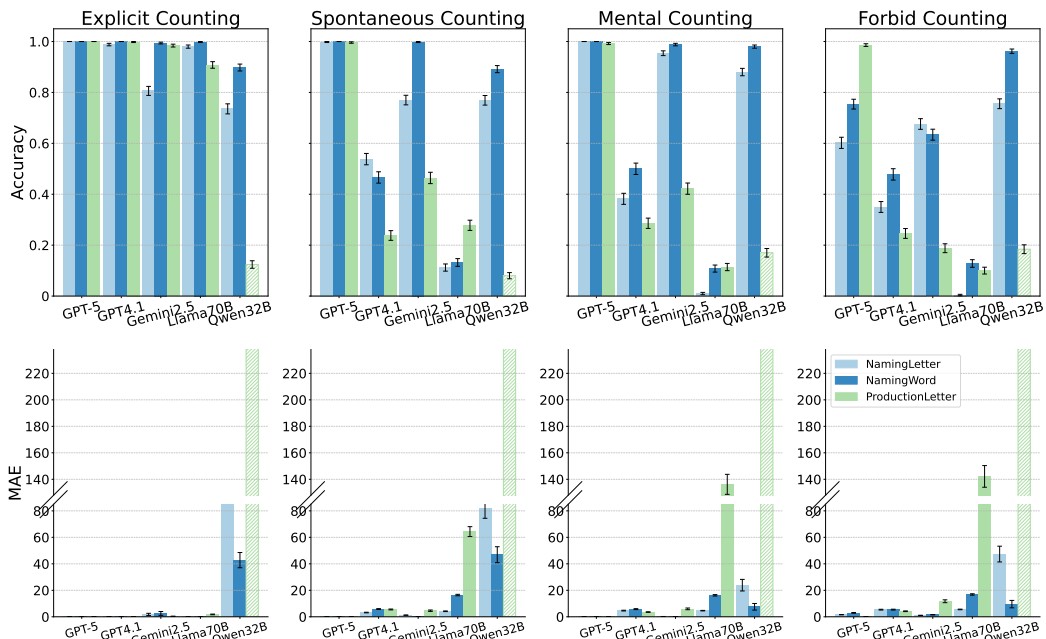

Figure 7: Comparison of accuracy and mean absolute error (MAE) across different models and prompting conditions on heterogeneous stimuli. Bars represent accuracy or MAE for each model (x-axis) across the tasks. Error bars indicate the variability of the estimates: for accuracy, binomial standard errors are shown; for MAE, standard errors of the mean are shown. We flagged Qwen's results as it has a large number of invalid trials.

### G.5 COUNTING VIA CODING

To investigate whether LLMs' coding ability could support perfect sequential enumeration, we select the SOTA coding LLM, GPT-5 as the testing model (Aider, 2025). As generating letters or words, either uniformly or non-uniformly, is too easy, we focus only on the non-uniform naming task. We gave 30 trials for GPT-5 to generate a code snippet that can count the number of letters or words in a given string. The prompt is: ″You need to write a code snippet to count how many letters or words in a given string.  The input is a string and the output of the snippet should be an integer number. Try your best to consider all possible conditions and return the final Python code.″ with the instruction (work as system message in other LLMs): ″You are a code expert!″.

After human inspection and testing the code snippets, our results revealed that only 11 trials (36%) are suitable for a string that could be either words or strings. Other code snippets require specification of whether the goal is to count words or letters. Among those valid trials, 10 trials returned perfect accuracy, and one code snippet achieved 50% of accuracy. We provide the generated code snippet in the drive folder.

### G.6 ANALYSIS OF EMBEDDINGS

Figure 8 illustrates the evolution of the first five PCs over the course of generation steps across the four counting conditions. Notably, the mental and forbid counting conditions exhibit highly similar dynamics across all five PCs, suggesting shared underlying representational structures but differences in the precision of those representations. In contrast, the explicit counting condition reveals distinct spikes at multiples of five and decade numbers, indicating a more discretized and externally guided numerosity representation. Although the spontaneous counting condition is behaviorally similar to the mental and forbid conditions, its latent dynamics display a hybrid pattern. Specifically, for some target numerosities, spontaneous counting exhibits internal accumulator-like patterns similar to the

mental and forbid conditions, while for others, it shows spike patterns aligned with those observed in the explicit condition. This suggests that spontaneous counting may engage a mixture of internal and externally anchored mechanisms in its representational trajectory.

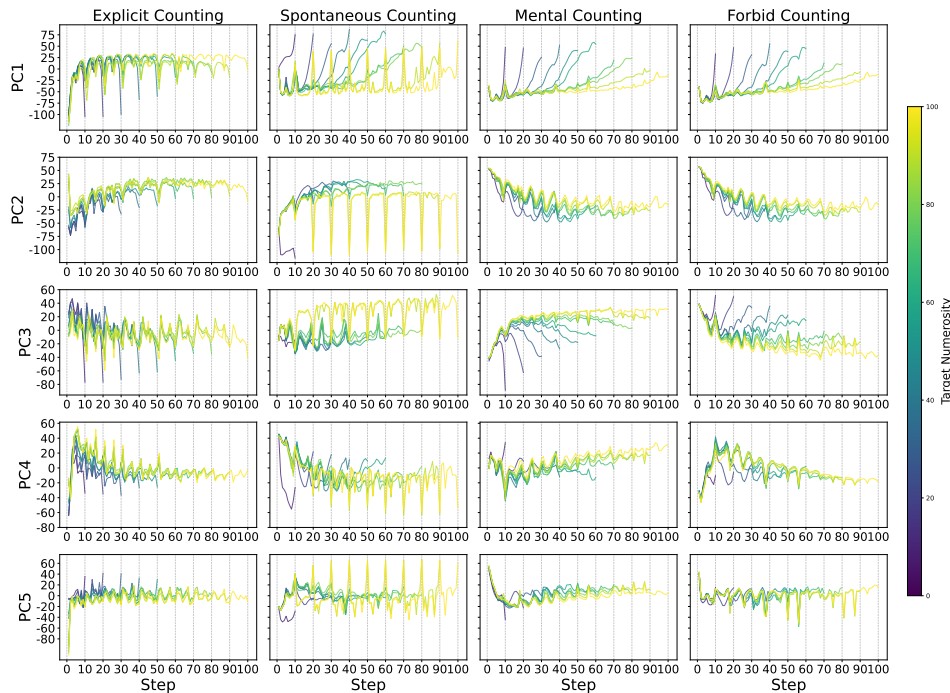

Figure 8: Trajectories of the first five principal components computed over hidden states for explicit condition only. Each subplot shows the trajectories along the principal components as a function of generation step, with colors indicating the target numerosity.

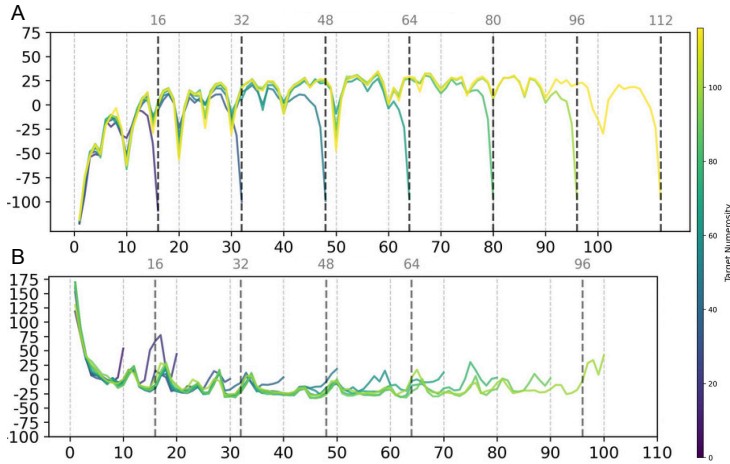

Figure 9: Trajectories of the first principal components computed over hidden states for explicit condition only. Panel (A) showed the hidden dynamics for counting in base-10 with target numbers being multiples of 16. Panel (B) showed the hidden dynamics for counting in base-16 with target numbers being multiples of 10.

### G.6.1 BASE-16 COUNTING

To better understand what drives the dips or spikes at the decade numbers in the explicit condition, a control experiment was conducted by asking the model to count explicitly in base-16 (0x00, 0x01, 0x02,...) with target numbers being multiples of 10 (*i.e.,* "decades" as 10,20,30...100), and to count explicitly in base-10 (1,2,3,...) with target numbers being multiples of 16 (16,32,48...112). The results show that the model is drawn by the multiples of 16 (see Fig 9 panel B for "spikes" of hidden dynamics on those steps), but finally the model spikes at the target number. Panel A further shows that the dips are present for both the target numbers and the multiples of 10.

### G.6.2 CORRELATION ANALYSIS OF POPULATION ENCODING

To quantify the relationship between individual neuron activations and sequence position, we computed Pearson correlation coefficients between each neuron's activation trajectory and the corresponding step indices across all trials in Section 4.3. Among the 8192 neurons, the top 500 with the strongest positive correlations exhibited a mean correlation coefficient of $0.523 \pm 0.003$ (SEM), while the top 500 with the strongest negative correlations had a mean of $-0.527 \pm 0.003$ (SEM). In both cases, the average p-value across neurons was $< .001$, indicating a highly significant association between neural activity and sequential position.

