# OpenReview forum: "Sequential Enumeration in Large Language Models"
_ICLR.cc/2026/Conference — ICLR 2026 Conference Withdrawn Submission_

### Official Review · Reviewer_XQ9p · 2025-10-29

**Soundness:** 2
**Presentation:** 2
**Contribution:** 1
**Rating:** 2
**Confidence:** 4

**Summary:**

In this paper, the authors study the capability of sequential enumeration of five different LLMs using two tasks: numerosity naming and numerosity production. In addition, the paper also studies whether an “internal counter mechanism” emerges in the model by looking into the latent space of them. Different prompting methods are used to observe the behaviors of LLMs. The paper also looks into the latent behavior of open-sourced LLMs.

**Strengths:**

- The paper studies a known yet interesting phenomenon of the counting ability of LLMs.
- Looking beyond accuracy to hidden-state patterns is a useful angle.

**Weaknesses:**

- Novelty concern: I don’t think the paper proposes anything new to the literature. The results (explicit > mental > spontaneous > forbid) feel expected. If the main takeaway is “use explicit counting prompts,” the contribution is limited without deeper analysis of why or any causal evidence.
- Word list transparency: The datasets are synthetic and rely on 5-letter, one-token words, but the exact list and counts are not provided. Please report: (i) how many candidate words were considered; (ii) which words passed per tokenizer; (iii) pass rates per model.
- It’s unclear if results hold for other word sets. An ablation with multiple resampled lists (and “frequent” vs “rare” words) would help rule out frequency bias.
- Several claims are suggestive but not tightly backed by analysis (e.g., “similar approximate estimation mechanisms,” “improve internal encoding”). These need statistics, layer-wise checks, or causal tests.
- Focus is on last-layer PCA for one open model. No layer-wise sweep (you could add a middle layer/ first layer for deeper analysis).
- Writing clarity: Some sections read unclear to me. For example, in Sec. 4.1, you make mechanism-level claims without showing supporting tests. I would suggest you reduce the amount of lengthy descriptions.

**Questions:**

- Why last-layer PCA only? Please add a layer-wise analysis (e.g., simple linear probe for step index with R² across layers) to see where the signal emerges.
- Spontaneous counting input: If no instruction is given, what exactly is the model’s input in naming vs production? Show a concrete example for each.
- Highlight task definitions: Move or emphasize Sec. 3.2 with short, concrete examples for “naming” and “production” so readers can follow later results.
- Word list release: please provide the full word list used in experiments, plus per-tokenizer pass tables and a script to reproduce the checks.
- Typos/format: L198 “<my\_answer>”

---

### Official Review · Reviewer_CLdR · 2025-10-30

**Soundness:** 2
**Presentation:** 3
**Contribution:** 2
**Rating:** 4
**Confidence:** 4

**Summary:**

This paper evaluates whether LLMs can count sequences of simple items. The authors test five models across two tasks, naming and production, under varied prompting strategies, and analyze internal neural dynamics to understand counting mechanisms.

**Strengths:**

It is a **well-posed question** and carefully designed **descriptive** analyses that examine whether current LLMs can count.

- tests multiple models, task types, and prompting setups.
- usings PCAs in the Lama model to reveal internal strategies for counting.
- ensures one-token-per-word to isolate counting from tokenization issues

**Weaknesses:**

Besides the strong descriptive analysis showing the limitations of LLMs, the study does not go deeper to investigate where the counting ability comes from. Different model architectures may encode numerical skills through distinct mechanisms, and performance may also reflect biases learned from training data frequency rather than systematic counting ability. Without separating these factors, the paper cannot fully explain how or why counting emerges in current models.

- some experimental design elements feel arbitrary, e.g., testing base-16 without a clear motivation.

- only the neural dynamics of the LLaMA model were studied, which did not exhibit high performance.

- prompt generation is not robust enough

**Questions:**

- Why is naming so much harder? Is it memory limitations, attention, or something else?

- So what if LLMs can't count spontaneously? Why does this matter beyond theory?

- How would you explain the performance difference between different models? Where does it come from? How can we quantify it?

- Would it be meaningful to design experiments with small language models to control the architecture and training data?

---

### Official Review · Reviewer_gNjx · 2025-10-31

**Soundness:** 3
**Presentation:** 4
**Contribution:** 3
**Rating:** 8
**Confidence:** 4

**Summary:**

This work studied the ability of LLMs to perform sequential enumeration. Structural settings were constructed to evaluate the ability of LLMs across various families to count elements and generate elements given a number.

**Strengths:**

- Experiments are well-designed and considered many aspects of sequential enumeration, with clear descriptions of tasks that cover many situations in sequential enumeration.
- Diverse types of Prompts (Explicit, Spontaneous, Mental, Forbid) were implemented and studied.
- The approach presented in sections 3.5, results in 4.2, and Figures 3,4 provided interesting analysis on the hidden states of the LLMs during enumeration.

**Weaknesses:**

Although the paper is mainly on evaluating the ability of the LLMs, it can be improved if a strategy to improve the ability related to sequential enumeration of the LLMs is discussed.

**Questions:**

In the author’s opinion, whether it is possible for an inexpensive approach (for example, steering approach) to improve the ability of open-source model in counting?

---

### Official Review · Reviewer_2cnc · 2025-10-31

**Soundness:** 3
**Presentation:** 3
**Contribution:** 2
**Rating:** 6
**Confidence:** 4

**Summary:**

This paper systematically investigates the sequential enumeration capabilities of state-of-the-art Large Language Models (LLMs). Through a series of naming (how-many) and production (give-N) tasks using various prompting strategies, the authors find that no tested LLM spontaneously employs a counting algorithm. While explicit instructions can elicit high accuracy, particularly in production tasks, the models struggle otherwise. A key finding from analyzing the internal dynamics of Llama-70B is that LLMs appear to use fundamentally different strategies: a superficial, token-based mechanism for explicit counting, versus a more continuous, "accumulator-like" dynamic for mental counting. The study concludes that LLMs still lack a robust, systematic understanding of enumeration, highlighting a significant gap between their emergent abilities and true procedural reasoning.

**Strengths:**

- Clarity: The paper is well-written, with a clear and logical structure.
- Significance:  Systematically demonstrating a critical failure mode in modern LLMs: the inability to spontaneously deploy a basic, procedural algorithm like counting.

**Weaknesses:**

- Familiar findings: The paper's main conclusion—that LLMs struggle with counting—confirms a known limitation, which may obscure the work's core novelty.
- The introduction does not show the paper's most compelling result.  The paper's key intellectual contribution is not apparent until late in the results section, weakening the narrative and its initial impact.
- Limited Scope of Mechanistic Analysis: PCA and neuron analysis is performed only on one model, Llama-70B. While this provides a fascinating case study, it is unclear if these findings generalize.
- Correlational Nature of Internal Analysis: The analysis of hidden states is correlational. While the PCA trajectories and neuron tuning curves are highly suggestive of an "internal counter," they do not establish causality.

**Questions:**

see weakness

---

### Official Review · Reviewer_6kNb · 2025-11-01

**Soundness:** 2
**Presentation:** 3
**Contribution:** 2
**Rating:** 2
**Confidence:** 4

**Summary:**

This paper tests models on tasks that require counting, such as counting the number of words in a sequence, or generating a sequence containing N words for some N. They find that differences in prompting, such as instructing the model to explicitly count or keep internal counts, greatly influences the model's success at such tasks.

**Strengths:**

This case study on counting demonstrates the importance of prompt instructions and that LLMs can be sensitive to instructions about how they should compute things internally. It also provides some insights into how LLMs keep track of counts of words.

**Weaknesses:**

The contributions of this paper ultimately seem somewhat narrow, as they are restricted to counting-based tasks. It is unclear why it is important to specifically understand LLM performance on these tasks. In addition, while the analysis of the model internals is interesting, it seems somewhat shallow and it's not clear what to take away. A strong conference paper should at least either have deep insights (even about a narrow task) or analyze a diverse set of important tasks (even if somewhat shallowly). As is, the paper seems more suitable for a workshop.

Some hypotheses do not seem very well-justified:
* Line 245: The fact that the gap reduces between the naming and production tasks is at best very weak evidence that these tasks use similar mechanisms, as the tasks are very different in nature and different mechanisms are possible.
* Line 320: It is hypothesized that Fourier features are involved in the computation. This seems like something that could be explicitly checked in open-weight models.
* Line 366: It's not clear how the figure suggests a "consistent underlying computational mechanism," or for that matter what this phrase means precisely.

I am not convinced by the use of PCA, as opposed to probes or logit lens to identify subspaces relevant to counting. Is it possible that there is a counting mechanism, but just that PCA does not surface it? Some sort of linear probe to extract the number of words seen would be able to identify the most relevant linear direction.

**Questions:**

Line 144-145: I'm confused about the decision to use increments of 10. In the Embers of Autoregression paper, they show that LLM success depends on the output probability, and multiples of 10 are more common than other numbers. This is one of their explicit examples https://arxiv.org/pdf/2309.13638 . By using increments of 10, the experiment seems to be ignoring this consideration. What was the motivation for using increments of 10?

Why perform PCA at the final layer, as opposed to intermediate layers? Intermediate layers may have richer features where counting happens.

When comparing the mental counting and explicit counting mechanisms, could you take the principal components identified from mental counting and see if those same directions have similar behavior in the explicit counting prompt? I am wondering if those directions have the same function in both conditions, but PCA returns different PC's.

---

### Note · Authors · 2025-11-25

**Comment:**

Thanks for the reviews.
We have other plans for the paper.

**Withdrawal Confirmation:**

I have read and agree with the venue's withdrawal policy on behalf of myself and my co-authors.